# Is UA/HDL-C a Reliable Surrogate Marker for Fatty Liver? A Comparative Evaluation with Metabolic Scores in a Mexican Population: The Genetics of Atherosclerotic Disease Study

**DOI:** 10.3390/diagnostics15111419

**Published:** 2025-06-03

**Authors:** Rosalinda Posadas-Sánchez, Giovanny Fuentevilla-Álvarez, Gilberto Vargas-Alarcón, Guillermo C. Cardoso-Saldaña

**Affiliations:** 1Endocrinology Department, Instiuto Nacional de Cardiologia Ignacio Chavez, Juan Badiano No. 1, Col. 4 Seccion XVI, Mexico City 14080, Mexico; rossy_posadas_s@yahoo.it (R.P.-S.); fuentevilla_alvarez@hotmail.com (G.F.-Á.); 2Department of Molecular Biology and Research Direction, Instiuto Nacional de Cardiologia Ignacio Chavez, Juan Badiano No. 1, Col. 4 Seccion XVI, Mexico City 14080, Mexico; gvargas63@yahoo.com

**Keywords:** UA/HDL ratio, fatty liver disease, index

## Abstract

**Background:** Fatty liver disease (FLD), the most common liver disease worldwide, is associated with cardiometabolic diseases and increases the risk of cardiovascular disease. It remains asymptomatic in its early stages, and late diagnosis heightens the likelihood of progression to severe liver diseases. **Objectives:** We aimed to evaluate the utility of serum uric acid to HDL cholesterol ratio (UA/HDL-C) as a biomarker for FLD and compare its diagnostic utility versus established liver disease index (FLI, LAP, HSI, NAFLD score (FLS), and ALT/AST ratio). **Methods:** This cross-sectional study, conducted between 2009 and 2013, included 1470 adults, 50.2% women and 49.8% men between 20 and 75 years old. FLD was diagnosed using non-contrast computed tomography. The population was stratified by sex and FLD. The associations with UA/HDL-C were analyzed using ROC curves and logistic regression analysis to evaluate and compare the predictive capacity of various indices for FLD. **Results:** Anthropometric, physiologic, biochemical variables, ratios, and indices were significantly higher in subjects with FLD (*p* < 0.001). In the unadjusted logistic regression model, UA/HDL-C is strongly associated with FLD (co-efficient 2.5, *p* < 0.001). The FLS, HSI, and ALT/AST ratios were also significant, whereas FLI and LAP showed no clear relationship. In the sex-adjusted model, the UA/HDL-C ratio remained strongly associated with FLD (3.47, *p* < 0.001). **Conclusions:** Our results suggest that the UA/HDL-C ratio is associated with FLD as an established liver disease index and may be a practical, useful marker for FLD. The results highlight its potential as a scrutiny and early biomarker for effective preventive strategies for FLD.

## 1. Introduction

Fatty liver disease (FLD) is the most common liver disease worldwide [1,2,3] and reaches 70–90% [4] in individuals with obesity [5] and type 2 diabetes [6]. FLD in Mexico is a significant public health issue with a prevalence exceeding 30% in the general population [2,7]. This liver disease increases the risk of cardiovascular disease [8], insulin resistance [9], and diabetes [10,11,12], making it a key component of metabolic syndrome [13]. Because it remains asymptomatic in early stages, diagnosis is often delayed, increasing the likelihood of progression to non-alcoholic steatohepatitis (NASH), fibrosis, cirrhosis, and hepatocellular carcinoma [14].

In Mexico, the high prevalence of obesity and diabetes has made FLD one of the leading causes of chronic liver disease [15,16]. Genetic predisposition also plays a role, with a high frequency of *PNPLA3* polymorphisms promoting lipid accumulation in the liver [17,18]. Additionally, diet and lifestyle factors, such as high consumption of refined sugars, contribute significantly to disease progression. These factors highlight the need for better risk prediction tools in this population [19].

FLD diagnosis relies on imaging techniques like ultrasound, elastography, and MRI, but these methods have limitations in cost, accessibility, and variability [20,21]. As a result, non-invasive metabolic indices, such as Fatty Liver Index (FLI) [22], Hepatic Steatosis Index (HSI) [23], Lipid Accumulation Product (LAP) [24], and NAFLD score (FLS) [25], have been developed. While these models perform well in different populations, their accuracy can vary based on ethnicity, sex, and metabolic characteristics.

Uric acid, a byproduct of purine metabolism, is linked to insulin resistance, systemic inflammation, and oxidative stress, which are all key factors in FLD [26]. Elevated levels are associated with visceral adiposity, endothelial dysfunction, and liver disease progression, whereas low HDL cholesterol is tied to metabolic syndrome and liver damage [27]. The UA/HDL-C ratio may reflect the balance between oxidative stress and antioxidant protection, making it a potential metabolic liver damage marker [28].

Recent studies suggest UA/HDL-C is an independent predictor of FLD, performing similarly to traditional indices; therefore, it has been proposed as a novel inflammatory and metabolic marker associated with increased FLD risk [29]. It was more predictive of the onset of FLD than UA or HDL-C alone [30]. This new biomarker may offer a practical and accessible alternative for detecting FLD in the general population.

Its role has not been well studied in Mexican populations, where genetics and the high prevalence of cardiometabolic risk factors, such as obesity, diabetes, and metabolic syndrome, may modulate the AU/HDL-C ratio predictive value. Additionally, sex differences in uric acid and lipid metabolism could influence its utility as a biomarker. Men tend to have higher uric acid levels, while women generally have higher HDL cholesterol, particularly before menopause [31]. Some evidence suggests that uric acid’s impact on FLD is more substantial in men, whereas in women, it may be influenced by estrogen levels and fat distribution [32]. However, further research is needed to determine whether UA/HDL-C ratio diagnostic accuracy differs by sex. UA/HDL-C could become a cost-effective tool for early detection, improving prevention and diagnosis strategies, particularly in populations with a high metabolic syndrome burden. This study aims to evaluate the UA/HDL-C ratio as a predictor of FLD in a Mexican cohort. It compares its diagnostic utility to established accepted and robust indices for liver disease (FLI, LAP, HSI, NAFLD score (FLS), and ALT/AST ratio).

## 2. Materials and Methods

### 2.1. Study Population

This cross-sectional study is nested within the Genetics of Atherosclerotic Disease (GEA) study, which investigates the associations between genetic polymorphisms and atherosclerosis in Mexican individuals. The GEA cohort was recruited between June 2008 and January 2013 at the National Institute of Cardiology. The present study included 1470 Mexican Mestizo volunteers from the GEA cohort who were enrolled following a medical evaluation and a health questionnaire. Figure 1 shows the main inclusion criteria were the absence of a personal or family history of coronary artery disease, no current or previous congestive heart failure, and the assessment of fatty liver disease status, which allowed for the classification of participants into two groups: individuals with and without FLD. Exclusion criteria included liver diseases other than fatty liver disease and renal, thyroid, and oncological diseases, determined through medical examination and clinical chemistry tests.

### 2.2. Computed Tomography Assessment

Non-contrast computed tomography (CT) is a validated method to quantify abdominal fat, including total abdominal fat (TAF), subcutaneous abdominal fat (SAF), and visceral abdominal fat (VAF) [32]. This study performed abdominal CT using a 64-channel detector helical tomography system (Somatom Sensation, Siemens, Malvern, PA, USA). To quantify TAF, SAF, and VAF, scans were obtained without contrast, as described by Kvist H. et al. [33].

### 2.3. Liver Fat Diagnosis

Non-alcoholic fatty liver disease was diagnosed using the Liver/Spleen Attenuation Ratio (L/SAR), with a ratio of <1.0 indicating the presence of FLD, as described by Longo R. et al. [32].

### 2.4. Cardiometabolic Risk Factors

Lifestyle factors, such as current smoking, high alcohol consumption (>20 g/d), and total physical activity (leisure, work, and exercise time), which may significantly influence FLD risk, were considered confounding variables [33]. To determine insulin resistance and excess adipose tissue, cut-off values were established based on a sample of 316 individuals (131 men and 185 women) who met the following criteria: body mass index (BMI) below 30 kg/m^2^, and no diagnosis of diabetes mellitus, dyslipidemia, or arterial hypertension.

Insulin resistance was defined as HOMA-IR > 3.44 in men and >3.45 in women. Arterial hypertension was diagnosed with systolic BP ≥ 140 mmHg, diastolic BP ≥ 90 mmHg, and/or antihypertensive treatment. Adipose tissue thresholds (75th percentile) were VAF ≥ 152.5 cm^2^ in men and ≥121 cm^2^ in women, SAF ≥ 221 cm^2^ in men and ≥320.5 cm^2^ in women. Overweight was classified as BMI of 25–29.9 kg/m^2^, and obesity as a BMI ≥ 30 kg/m^2^. Dyslipidemia, based on NCEP-ATP III criteria, included hypercholesterolemia (TC ≥ 200 mg/dL or LDL-c ≥ 160 mg/dL), hypertriglyceridemia (TG ≥ 150 mg/dL), and hypoalphalipoproteinemia (HDL-c < 40 mg/dL in men or <50 mg/dL in women). Metabolic syndrome, also based on NCEP-ATP III criteria, required central obesity (waist ≥ 90 cm in men, ≥80 cm in women) plus at least two of the following: TG > 150 mg/dL or treatment, HDL-C < 40 mg/dL in men or <50 mg/dL in women, BP > 130/85 mmHg or treatment, and fasting glucose ≥ 126 mg/dL.

### 2.5. Laboratory Analyses

Following a 10 h fasting period and 20 min seated position, venous blood samples were drawn into tubes with and without K2 EDTA anticoagulant (1.8 mg/mL). Plasma was used to assess glucose, total cholesterol (TC), triglycerides (TG), and high-density lipoprotein cholesterol (HDL-C) levels. Serum samples were analyzed for alanine aminotransferase (ALT), aspartate aminotransferase (AST), gamma-glutamyl transferase (GGT), and uric acid.

All biochemical analyses were completed within three days of sample collection. The measurements were conducted using a Hitachi 902 autoanalyzer (Hitachi Ltd., Tokyo, Japan) with enzymatic colorimetric reagents (Roche/Hitachi, Mannheim, Germany). To ensure precision and consistency, lipid and lipoprotein quantifications were regularly validated through the Lipid Standardization Program of the Centers for Disease Control and Prevention (LSP-CDC, Atlanta, GA, USA). The intra- and inter-assay variability remained below 3% across all analyses.

### 2.6. Comparison of Indices for Predicting FLD

We compared several established indices to predict the presence of FLD against the UA/HDL-C ratio. The selected indices were the Fatty Liver Index (FLI) [34], NAFLD score (FLS) [35], Lipid Accumulation Product (LAP) [36], and Hepatic Steatosis Index (HSI) [37]. These indices were chosen for their ability to predict liver fat accumulation using different combinations of clinical, biochemical, and demographic data. The formulas used to calculate these indices were as follows:

**1.** Fatty liver index (FLI):
$$\mathrm{FLI}=\frac{e^{0.953\times \log\left(\mathrm{Triglycerides}\right)\times (0.139\times \mathrm{BMI})+0.718\times \log\left(\mathrm{GGT}\right)+0.053\times \mathrm{Waist}\;\mathrm{Circumference}-15.745}}{1+e^{0.953\times \log\left(\mathrm{Triglycerides}\right)\times (0.139\times \mathrm{BMI})+0.718\times \log\left(\mathrm{GGT}\right)+0.053\times \mathrm{Waist}\;\mathrm{Circumference}-15.745}}\times 100$$**2.** NAFLD score (FLS):
$$\begin{array}{cc} \mathrm{NAFLD}\;\mathrm{Score}= & -2.89+\left(1.18\times \mathrm{Metabolic}\;\mathrm{Syndrome}\right)+\left(0.45\times \mathrm{DMII}\right)\\ & +\left(0.15\times \mathrm{BMI}\right)-\left(0.04\mathrm{Age}\right)+(1.13\times \frac{\mathrm{AST}}{\mathrm{ALT}})+(0.94\times \mathrm{ApoA}1) \end{array}$$**3.** Lipid Accumulation Product (LAP):
$$\mathrm{LAP}=\left(\mathrm{Waist}\;\mathrm{circumference}-\mathrm{reference}\;\mathrm{value}\right)\times \mathrm{triglycerides}$$**4.** Hepatic Steatosis Index (HSI):
$$\mathrm{HSI}=8\times \frac{\mathrm{ALT}}{\mathrm{AST}}+\mathrm{BMI}+2(\mathrm{Sex})$$

### 2.7. Statistical Analysis

The dataset contained variables related to lipid profile, glucose, insulin, liver function, adiposity, and demographic characteristics, which underwent preprocessing before statistical analysis. The primary outcome was FLD diagnosis, while sex was included as a covariate. To ensure comparability across indices, all variables were normalized using Min-Max scaling. Analyses were conducted for the total study population and stratified by sex to evaluate potential sex-specific differences in predictive capacity. We conducted a receiver operating characteristic (ROC) curve analysis to evaluate and compare the predictive capacity of various indices for FLD. The metabolic indices included the FLI, FLS, LAP, HSI, and the UA/HDL-C ratio. To enhance predictive accuracy, we developed a composite index by integrating the probability scores of each index using a multivariable logistic regression model. The area under the curve (AUC), 95% confidence intervals (CI), and the Youden Index were calculated to assess the discriminative ability of each predictor. The Youden Index was the maximum (sensitivity + specificity − 1). Analyses were stratified by sex to explore potential differences in predictive performance. Logistic regression models were constructed to further evaluate the association between each metabolic index. Both unadjusted and adjusted models were tested, with the adjusted models, including sex as a covariate. Results were reported as odds ratios with 95% confidence intervals. Statistical significance was set at *p* < 0.05. All analyses were performed using Python V.3.12.

### 2.8. Protein–Protein Interaction Network Analysis

To explore the molecular mechanisms linking hyperuricemia and hypoalphalipoproteinemia to fatty liver disease (FLD), we performed a protein–protein interaction (PPI) analysis. Genes associated with elevated uric acid and reduced HDL-C levels were identified based on literature reports and metabolic pathway relevance. The gene set was input into ShinyGO v0.82 [38] using *Homo sapiens* as the reference organism. Functional enrichment and interaction networks were generated using the Kyoto Encyclopedia of Genes and Genomes (KEGG) [39] database to visualize signaling pathways involved in lipid metabolism, inflammation, and liver damage. The resulting network illustrated connections between metabolic stressors, inflammatory cascades, and hepatic outcomes relevant to FLD pathogenesis.

### 2.9. Ethical Statement

Participants provided written informed consent. The study complied with the Declaration of Helsinki. The National Institute of Cardiology Ignacio Chávez Research Committee approved the project (protocol number 09-646).

## 3. Results

### 3.1. Population Characteristics

The study included 1470 adult individuals, 982 (66.8%) without fatty liver disease (FLD) and 488 (33.2%) with FLD. Among those without FLD, 518 (52.7%) were women and 464 (47.3%) were men. In the FLD group, 220 (45.1%) were women and 268 (54.9%) were men. The median age was similar across groups, ranging from 52.5 to 54 years. Individuals with FL had a higher BMI compared to controls. Women had higher HDL-C levels than men in both groups. Among controls, the median was 51.3 mg/dL in women and 41.0 mg/dL in men. In individuals with fatty liver, HDL-C decreased to 45.6 mg/dL in women and 38.6 mg/dL in men (*p* < 0.001). Uric acid levels were higher in men across all groups. In controls, the median was 6.12 mg/dL in men and 4.62 mg/dL in women. In the fatty liver group, levels increased to 6.76 mg/dL and 5.23 mg/dL in women, respectively (*p* < 0.001). Additionally, we compared several predictive indices for fatty liver disease, the FLI, NAFLD Score, LAP, and HSI, and biochemical ratios, including the UA/HDL-C ratio, as an additional marker. All predictive indices were higher in subjects with FLD, *p* < 0.001. Table 1

### 3.2. Comparison of Ratios and Indices for Predicting FLD

We aimed to identify the most accurate predictor of FLD by assessing its diagnostic performance. As shown in Figure 2, all indices demonstrated significant predictive ability for fatty liver disease (*p* < 0.001). The FLI achieved the highest area under the curve (AUC = 0.755) and the highest Youden Index (0.35), indicating the best overall performance. The UA/HDL-C ratio showed a lower AUC (0.679) and a smaller Youden Index (0.23), reflecting a more limited diagnostic utility. The optimal cut-off points were 7.39 for FLI (sensitivity = 0.80, specificity = 0.55) and 0.12 for UA/HDL-C ratio (sensitivity = 0.66, specificity = 0.57).

We developed a composite index by integrating the individual indices using a multivariable logistic regression model to improve the predictive accuracy for fatty liver disease. This new index demonstrated the highest discriminative ability, with an AUC of 0.763 and a Youden Index of 0.37, surpassing the performance of any single index predictor.

Due to the simplicity of the UA/HDL-C ratio, which is derived from only two parameters, its comparison against well-established composite indices may not fully reflect its clinical utility. Therefore, we compared the UA/HDL-C ratio with other individual biochemical ratios proposed as surrogate markers for fatty liver disease to assess its performance better. This analysis was stratified by sex to account for the known differences in HDL-C levels between men and women in the Mexican population.

We performed a sex-stratified ROC curve analysis to determine their discriminative ability and compared different biochemical ratios for developing FLD. Beyond established predictive indices, we also evaluated additional metabolic biochemical ratios (LDL-C, ALT/AST, and LDL-C/HDL-C ratio), as shown in Figure 3. The UA/HDL-C ratio demonstrated the best performance, particularly in men (AUC = 0.687, *p* < 0.001), highlighting its substantial predictive value in this group. In women, it also showed one of the highest AUCs (0.622, *p* < 0.001). In both populations, it outperformed uric acid (AUC = 0.655 in men, 0.618 in women) and the ALT/AST ratio (AUC = 0.676 in men, 0.659 in women). LDL-C had no discriminative power, while the LDL-C/HDL-C ratio showed moderate performance in women (AUC = 0.554, *p* = 0.015), as shown in Figure 3. These results reinforce the UA/HDL-C ratio as a key marker for fatty liver disease assessment, particularly in men.

Table 2 shows that the UA/HDL-C ratio showed the strongest association with fatty liver in the unadjusted logistic regression model, with the highest coefficient and a highly significant *p*-value. The NAFLD Score, HSI, and ALT/AST ratio also showed significant associations, while FLI and LAP did not demonstrate a clear relationship with fatty liver. The UA/HDL-C ratio remained the most strongly associated predictor in the sex-adjusted model, increasing its coefficient. The NAFLD Score, HSI, and ALT/AST ratio maintained their significance, while FLI and LAP showed no significant association.

We performed Spearman correlations to assess the associations between the calculated indices and biochemical and anthropometric variables, stratifying the results by sex. Women with FLD reveal significant associations between metabolic indices and biochemical and anthropometric variables. Notably, the UA/HDL-C ratio correlates with glucose (*p* = 0.001), ALT (*p* = 0.008), and HOMA-IR (*p* < 0.001), highlighting its strong link to insulin resistance and liver metabolism. The UA/HDL-C ratio also shows associations with triglycerides and waist circumference, reinforcing its connection to metabolic syndrome. In men with FLD, the UA/HDL-C ratio is significantly correlated with uric acid (*p* < 0.001) but not with ALT (*p* = 0.058), nor with HOMA-IR (*p* = 0.117). In both sexes, UA/HDL-C follows a similar pattern to FLI, LAP, and HSI index, associating with key liver and lipid metabolism markers. However, its stronger correlation with ALT and HOMA-IR in women underscores its potential as a marker of hepatic dysfunction and insulin resistance in this group. (Appendix A).

## 4. Discussion

Our findings highlight the significant association between the UA/HDL-C ratio and FLD in men and women from the Mexican population and support that elevated uric acid and reduced HDL-C levels, together with insulin resistance, low inflammatory state, and visceral adiposity, have been extensively linked to FLD pathogenesis, reinforcing their role as key metabolic contributors to hepatic dysfunction. The UA/HDL-C ratio compared against accepted and robust indices to identify FLD shows that all indices perform superiorly to the UA/HDL-C ratio. While our results confirm that the UA/HDL-C ratio has a lower area under the curve (AUC) compared to traditional composite indices, such as the Fatty Liver Index (FLI), NAFLD score, and Hepatic Steatosis Index (HSI), it is important to consider the clinical context and utility of each approach. Composite indices integrate multiple anthropometric and biochemical parameters, enhancing diagnostic performance but requiring broader data availability and more complex calculations. In contrast, the UA/HDL-C ratio is derived from only two routinely measured laboratory parameters, making it highly accessible, cost-effective, and easy to implement in clinical settings, particularly in resource-limited environments.

Moreover, our findings suggest that while composite indices may be more robust in overall predictive capacity, the UA/HDL-C ratio provides complementary information that reflects specific metabolic disturbances—namely, hyperuricemia and hypoalphalipoproteinemia—both of which are independently linked to insulin resistance and hepatic steatosis. The sex-stratified analyses revealed that the UA/HDL-C ratio maintains the performance in men, highlighting its potential as a sex-sensitive marker. These findings support the combined use of composite indices for comprehensive evaluation and the UA/HDL-C ratio for initial screening or follow-up monitoring. The relatively lower performance of the UA/HDL-C ratio compared to these indices can be attributed to the fact that traditional indices incorporate multiple biochemical and anthropometric variables, making them more robust in detecting FLD. In contrast, UA/HDL-C is a more straightforward metric that captures key metabolic disturbances but may lack some of the broader predictive capacity of composite indices.

Recognizing this limitation, we subsequently compared UA/HDL-C against individual biochemical parameters associated with FLD, including LDL-C, uric acid, the ALT/AST ratio, and the LDL-C/HDL-C ratio, stratified by sex. In this context, UA/HDL-C emerged as the most effective marker, outperforming other biochemical indicators, particularly in men. The strong association in men may be linked to their inherently higher uric acid levels [40,41] and lower HDL-C concentrations [42], factors that predispose them to increased hepatic lipid accumulation.

Interestingly, the ALT/AST ratio, traditionally used as a liver injury marker [43], performed comparably but not superior to the UA/HDL-C ratio in either sex. This could reflect that ALT and AST elevations typically indicate more advanced hepatic damage [44], whereas the UA/HDL-C ratio may capture earlier metabolic alterations linked to steatosis onset. LDL-C and LDL-C/HDL-C ratios, despite their widespread use in cardiovascular risk prediction, showed weak to moderate discriminative power for FLD, reinforcing the notion that hepatic steatosis is more tightly linked to triglyceride-rich lipoprotein metabolism and insulin resistance than to isolated elevations in LDL-C.

Studies in Asian [45,46,47] cohorts have similarly reported an association between UA/HDL-C and metabolic syndrome components, reinforcing its potential as a metabolic risk marker. However, the degree of association may vary due to genetic predisposition, dietary habits, and environmental factors. The Mexican population has a high prevalence of obesity and insulin resistance [48,49], which could amplify the impact of hyperuricemia and dyslipidemia on hepatic steatosis [16].

In the present study, the logistic regression analysis further reinforced the significance of UA/HDL-C in FLD prediction. In the sex-adjusted model, UA/HDL-C exhibited the strongest association with FLD, with the highest coefficient and significance, suggesting that the UA/HDL-C ratio alone encapsulates key metabolic disturbances contributing to FLD. Men appear more susceptible to these metabolic disturbances due to sex-specific lipid and uric acid metabolism differences. Testosterone has been associated with lower HDL-C levels [50], while estrogen enhances HDL synthesis [51] and uric acid excretion [52]. This hormonal disparity may explain the stronger association between UA/HDL-C and FLD in men. Additionally, men generally have higher visceral adiposity, which contributes to increased uric acid production via the activation of xanthine oxidase and higher levels of pro-inflammatory cytokines, exacerbating FLD risk [53].

The strong association between the UA/HDL-C ratio and fatty liver disease (FLD) reflects its close relationship with key metabolic disturbances, such as insulin resistance, hepatic dysfunction, and dyslipidemia. Spearman correlation analysis revealed that in individuals with FLD, the UA/HDL-C ratio was significantly associated with glucose, alanine aminotransferase (ALT), and HOMA-IR, particularly in women, underscoring its link with impaired glucose metabolism and hepatic injury. Moreover, the ratio showed positive correlations with triglyceride levels and waist circumference, further highlighting its association with components of metabolic syndrome. These findings are consistent with previous reports that identify insulin resistance as a central driver of FLD pathogenesis and support using the UA/HDL-C ratio as a clinically accessible surrogate marker of metabolic dysfunction [54].

We conducted an in silico protein–protein interaction (PPI) analysis using ShinyGO v0.82 and KEGG pathway enrichment to explore the biological plausibility of these clinical associations at the molecular level. Genes previously implicated in hyperuricemia and hypoalphalipoproteinemia were used as input to construct an interaction network (Appendix A). The resulting analysis identified multiple signaling pathways involved in hepatic lipid accumulation, inflammation, and cellular stress response, which are relevant to FLD pathogenesis.

The insulin signaling pathway is prominently represented at the top of the interaction network. Activation of the insulin receptor (INSR) initiates downstream signaling through PI3K and Akt [26], which regulate the expression of key lipogenic transcription factors, such as SREBP-1c, promoting fatty acid synthesis and contributing to hepatic triglyceride accumulation [55]. In parallel, elevated uric acid levels and circulating free fatty acids (FFAs) trigger endoplasmic reticulum stress, activating stress-responsive proteins, such as XBP1 and TRAF2 [56]. These, in turn, activate pro-inflammatory cascades mediated by JNK1 and NF-κB [57], resulting in the production of inflammatory cytokines (IL-1, IL-6, TNF-α) and the induction of hepatocyte apoptosis [58].

This coordinated disruption of insulin signaling, lipid homeostasis, and inflammatory pathways provides mechanistic support for the clinical utility of the UA/HDL-C ratio in identifying individuals at risk for FLD. As illustrated in Appendix A, this ratio integrates two critical metabolic alterations—hyperuricemia and hypoalphalipoproteinemia—that converge on key molecular mechanisms driving hepatic steatosis, steatohepatitis, and fibrosis.

## 5. Limitations

The study’s cross-sectional nature precludes causal inferences between the UA/HDL-C ratio and the development of fatty liver disease. Longitudinal studies are needed to confirm whether alterations in this ratio precede hepatic steatosis or result from it. Another limitation of this work is that since the sample consisted of volunteers, participants may not have represented the general population. Nevertheless, the prevalence of coronary heart disease risk factors observed is similar to that found in the ENSANUT, a survey with national representation [59].

## 6. Conclusions

Our findings suggest that the UA/HDL-C ratio may be a metabolic indicator for FLD, associated with biochemical disturbances in hepatic lipid accumulation. While its predictive capacity is lower than traditional indices, its association with metabolic dysfunction suggests it could be a helpful marker. However, further studies are warranted to explore its potential utility in risk stratification and early detection of FLD.

## Figures and Tables

**Figure 1 diagnostics-15-01419-f001:**
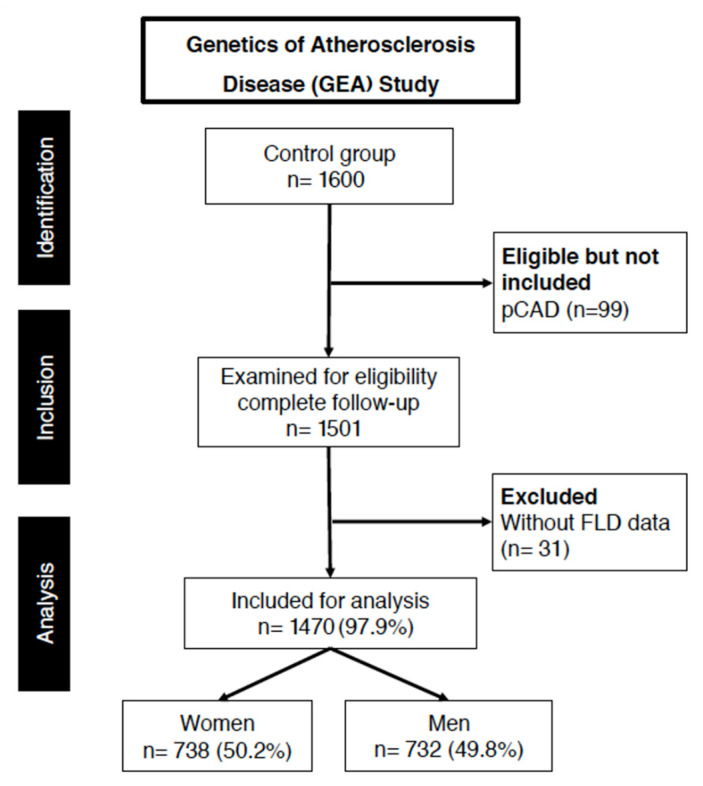
The participant flow diagram for the present analysis is derived from the GEA (Genetics of Atherosclerosis) Study. After excluding individuals with premature coronary artery disease (pCAD) and those lacking hepatic imaging data, a final sample of 1470 subjects was analyzed. The cohort was evenly distributed by sex, allowing for sex-stratified analyses of biochemical and metabolic markers associated with fatty liver disease (FLD).

**Figure 2 diagnostics-15-01419-f002:**
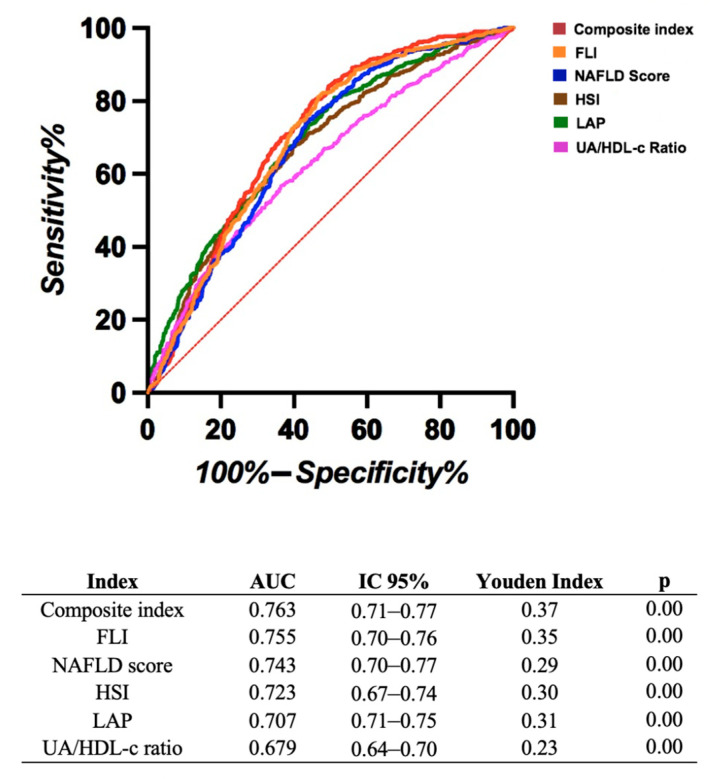
ROC curve analysis comparing indices for fatty liver disease, Fatty Liver Index (FLI), NAFLD score, Hepatic Steatosis Index (HSI), Lipid Accumulation Product (LAP), and the UA/HDL-C ratio. The Youden Index was calculated as the sum of sensitivity and specificity minus 1, representing the optimal cut-off for each index.

**Figure 3 diagnostics-15-01419-f003:**
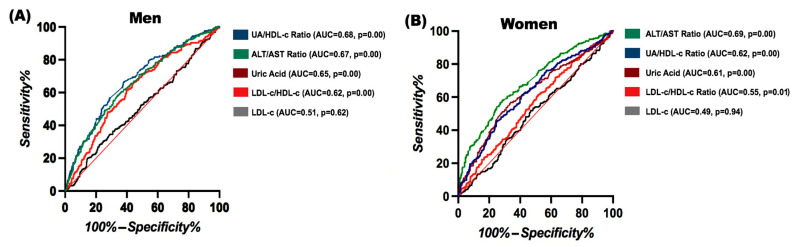
ROC curves comparing the ability of different metabolic parameters to discriminate fatty liver disease in men (**A**) and women (**B**). The curves represent the performance of LDL-C, uric acid, the ALT/AST ratio, the UA/HDL-C ratio, and the LDL-C/HDL-C ratio. AUC: area under the curve.

**Table 1 diagnostics-15-01419-t001:** Anthropometric and biochemical characteristics by sex between control and fatty fiver group.

Variable	Female NO FLD n = 518	Male NO FLD n = 464	Female FLD n = 220	Male FLD n = 268	p1	p2
Age (years)	54.00 (29.00–77.00)	54.00 (26.00–81.00)	53.00 (34.00–74.00)	52.50 (29.00–75.00)	0.54	0.33
BMI (kg/m^2^)	26.99 (17.26–39.84)	27.07 (18.93–39.90)	29.71 (20.33–39.86)	29.59 (19.58–39.84)	<0.001	<0.001
WC (cm)	88.20 (63.50–117.00)	95.20 (73.20–132.30)	95.65 (73.00–123.40)	101.50 (76.40–130.80)	<0.001	<0.001
DBP (mmHg)	71.00 (44.00–160.00)	78.00 (55.00–174.00)	73.50 (53.00–138.00)	77.00 (55.00–158.00)	<0.001	<0.001
SBP (mmHg)	108.00 (15.00–199.00)	116.00 (87.00–199.00)	111.50 (89.00–188.00)	119.00 (95.00–181.00)	<0.001	<0.001
HDL-c (mg/dL)	51.30 (22.00–97.60)	41.00 (18.60–76.40)	45.58 (22.36–100.80)	38.61 (16.88–97.50)	<0.001	<0.001
LDL-c (mg/dL)	116.06 (39.54–264.56)	117.99 (18.52–280.07)	117.93 (37.30–273.74)	120.96 (45.21–233.45)	0.55	0.95
Tg (mg/dL)	130.30 (40.00–540.10)	151.25 (41.00–581.00)	163.30 (50.00–735.00)	177.65 (54.90–682.50)	<0.001	<0.001
Glucose (mg/dL)	87.00 (58.00–391.00)	90.00 (72.00–324.00)	95.00 (62.00–271.00)	95.00 (71.00–295.00)	<0.001	<0.001
HOMA-IR	3.33 (0.22–27.08)	3.30 (0.44–26.44)	5.49 (1.07–22.91)	5.34 (1.46–34.11)	<0.001	<0.001
C-RP (mg/L)	1.50 (0.12–52.00)	1.11 (0.15–124.00)	3.06 (0.15–16.90)	1.67 (0.15–14.50)	<0.001	<0.001
VAF (cm^2^)	131 (23–446)	168 (25–504)	166 (53–389)	208 (48–459)	<0.001	<0.001
Uric Acid (mg/dL)	4.62 (2.07–9.20)	6.12 (2.70–11.72)	5.23 (2.36–8.68)	6.76 (2.82–11.03)	<0.001	<0.001
Physical activity index	7.85 (4.5–11.2)	7.81 (4.5–12.2)	7.82 (4.2–11)	8.1 (5.5–10.5)	-	-
AST (U/L)	22.00 (10.00–148.00)	25.00 (11.00–78.00)	27.00 (13.00–92.00)	29.00 (12.00–114.00)	<0.001	<0.001
ALT (U/L)	19.00 (4.00–220.00)	23.00 (5.00–125.00)	28.00 (8.00–151.00)	35.00 (6.00–145.00)	<0.001	<0.001
GGT (U/L)	20.00 (4.00–315.00)	29.00 (7.00–363.00)	27.00 (9.00–607.00)	40.00 (6.90–286.00)	<0.001	<0.001
Ratios and Fatty liver Index						
UA/HDL-C Ratio	0.09 (0.03–0.36)	0.15 (0.05–0.47)	0.12 (0.05–0.28)	0.17 (0.06–0.52)	<0.001	<0.001
ALT/AST Ratio	0.89 (0.22–2.31)	1.00 (0.23–3.33)	1.07 (0.38–2.75)	1.24 (0.26–3.63)	<0.001	<0.001
LDL/HDL Ratio	2.23 (0.01–9.80)	2.83 (0.66–7.04)	2.72 (0.47–6.75)	3.05 (0.92–6.87)	<0.001	<0.001
NAFLD Score	10.57 (6.47–53.53)	11.60 (8.22–45.20)	11.79 (6.80–37.75)	12.73 (8.73–41.26)	<0.001	<0.001
FLI	4.72 (0.43–100.00)	9.12 (0.50–100.00)	12.71 (0.59–100.00)	23.40 (0.45–100.00)	<0.001	<0.001
LAP	34.26 (0.00–240.77)	64.93 (12.86–307.49)	55.27 (7.88–215.76)	89.19 (15.54–464.66)	<0.001	<0.001
HSI	36.89 (22.72–58.65)	49.49 (34.71–69.88)	43.89 (26.40–59.28)	55.22 (39.99–69.80)	<0.001	<0.001
Comorbidities						
MS % (n)	11.5 (60)	57 (269)	29 (64)	79 (214)	-	-
Smoking % (n)	18.7 (97)	25.4 (118)	20.4 (45)	23.1 (62)	-	-
Alcohol consuption % (n)	0.4 (2)	2.8 (13)	0.5 (1)	4.9 (13)	-	-
HTA % (n)	7.1 (37)	13.3 (62)	8.1 (18)	13.4 (36)	-	-
T2DM % (n)	9.2 (48)	11.4 (53)	18.1 (40)	19 (51)	-	-

BMI: Body mass index; WC: Waist circumference; Tg: Triglycerides; HOMA-IR: Homeostatic Model Assessment for Insulin Resistance; C-RP: C-Reactive Protein; VAF: Visceral abdominal fat.; AST: Aspartate Aminotransferase; ALT: Alanine Aminotransferase; GGT: Gamma-Glutamyl Transferase; DBP: Diastolic Blood Pressure; SBP: Systolic Blood Pressure; FLI: Fatty Liver Index; LAP: Lipid Accumulation Product; HSI: Hepatic Steatosis Index; MS: Metabolic syndrome; HTA: Hypertension; T2DM: Type 2 Diabetes Mellitus. p1: Female Control vs. Female Fatty liver, p2: Male Control vs. Male Fatty liver. The data are expressed as median (Min-Max), statistical test: Mann–Whitney U test.

**Table 2 diagnostics-15-01419-t002:** Logistic regression analysis for fatty liver disease and related biomarkers.

Unadjusted Logistic Regression				
Variable	Coefficient	Standard Error	95% CI	*p*
UA/HDL-c Ratio	2.50	0.66	3.31–44.98	<0.001
FLI	0.44	0.27	0.90–2.70	<0.10
NAFLD Score	2.00	0.57	2.42–22.81	<0.001
LAP	0.13	0.80	0.23–5.54	0.86
HSI	1.16	0.46	1.29–7.88	0.01
ALT/AST Ratio	3.85	1.17	4.71–470.41	<0.001
Adjusted Logistic Regression (Sex)				
Variable	Coefficient	Standard Error	95% CI	*p*
UA/HDL-C ratio	3.47	0.70	8.18–128.18	<0.001
FLI	0.29	0.28	0.77–2.33	0.29
NAFLD Score	1.942	0.58	2.23–21.75	<0.001
LAP	−0.08	0.81	0.18–4.57	0.92
HSI	3.20	0.57	7.96–75.86	<0.001
ALT/AST Ratio	3.28	1.14	2.79–253.23	<0.001

Fatty Liver Index (FLI), NAFLD Score, Lipid Accumulation Product (LAP), and Hepatic Steatosis Index (HSI).

## Data Availability

Due to confidentiality agreements, the data underlying this study are not publicly available. Access to the data can be requested through gccardosos@yahoo.com following their confidentiality protocols.

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
