# Peer review of "Is UA/HDL-C a Reliable Surrogate Marker for Fatty Liver? A Comparative Evaluation with Metabolic Scores in a Mexican Population: The Genetics of Atherosclerotic Disease Study"

_diagnostics, 2025, doi:10.3390/diagnostics15111419_

Round 1
Reviewer 1 Report
Comments and Suggestions for Authors
Hello!
Congratulations on the article!
This article's scientific contribution is valuable, as it addresses a current topic and proposes a practical solution for the early detection of a frequently underdiagnosed disease.
Comments:
- At the end of the Introduction chapter, the authors should try to achieve a clearer separation between the purpose of the study and its justification (lines 78-83).
- The Conclusions chapter should be identified as such, so I advise you to mention it separately: 5. Conclusions (line 378).
Good luck!
Author Response
Dear Reviewer 1
Thank you for taking the time to review and comment on our manuscript,
Is UA/HDL-C a Reliable Surrogate Marker for Fatty Liver? A Comparative Evaluation with Metabolic Scores in a Mexican Population. (Diagnostics 3633690).
We found the advice constructive and incorporated many suggestions into our reviewed version.
We have responded to each comment individually and would like to draw your attention to the changes in the manuscript, highlighted in red.
Thank you again for your thoughtful comments.
Sincerely,
Dr. Guillermo C. Cardoso-Saldaña.
Comments:
- At the end of the Introduction chapter, the authors should try to achieve a clearer separation between the purpose of the study and its justification (lines 78-83).
Response:
Thanks for the suggestion. We restructured the last paragraph at the end of the introduction and described the objective separately. Lines 78-83
- The Conclusions chapter should be identified as such, so I advise you to mention it separately: 5. Conclusions (line 378).
Response:
Thanks for the suggestion; the conclusions were mentioned separately and we added Section 6 for better clarity. Conclusions. Lines 422-428
Reviewer 2 Report
Comments and Suggestions for Authors
Dear Author (s),
Thank you for submitting your work. Your study provides a systematic investigation into the relationship between the UA/HDL-C ratio and fatty liver disease (FLD), offering novelty and potential clinical value. However, to further enhance the scientific rigor and presentation quality of the manuscript, I recommend addressing the following points during revision:
-
Lifestyle factors such as smoking, alcohol consumption, and physical activity, which may significantly influence FLD risk, were not considered. Please clarify this aspect in the Methods or Discussion, or explicitly acknowledge it as a limitation.
-
Potential selection bias related to exclusion criteria and sample selection was not discussed. Please address possible sources of selection bias and their potential impact on the findings.
-
Figure formatting needs improvement; for example, axis labels in the ROC curves (“1–Specificity”, “Sensitivity”) should be capitalized according to international standards.
-
The ROC curves are thin, pale, and display aliasing, which compromises the clarity of key results. Please enhance figure quality by thickening lines and optimizing color schemes.
-
The best cut-off points, along with corresponding sensitivity and specificity values, are not indicated for major curves (e.g., the best model and the UA/HDL-C ratio). Including this information would strengthen the clinical relevance of the findings.
-
The Discussion lacks a comprehensive analysis of study limitations, particularly regarding the inability of cross-sectional studies to infer causality, sample representativeness, and generalizability issues. Please add a structured limitations section.
-
The manuscript does not sufficiently compare the UA/HDL-C ratio with traditional composite indices (e.g., FLI, LAP) in terms of predictive ability and practical application. A more detailed discussion highlighting their complementary roles would be beneficial.
Author Response
Thank you for taking the time to review and comment on our manuscript, Is UA/HDL-C a Reliable Surrogate Marker for Fatty Liver? A Comparative Evaluation with Metabolic Scores in a Mexican Population. (Diagnostics 3633690).
We found the advice constructive and incorporated many suggestions into our reviewed version.
We have responded to each comment individually and would like to draw your attention to the changes in the manuscript, highlighted in red.
Thank you again for your thoughtful comments.
Sincerely,
Dr. Guillermo C. Cardoso-Saldaña.
Reviewer 2
Thank you for submitting your work. Your study provides a systematic investigation into the relationship between the UA/HDL-C ratio and fatty liver disease (FLD), offering novelty and potential clinical value. However, to further enhance the scientific rigor and presentation quality of the manuscript, I recommend addressing the following points during revision:
Response:
We sincerely thank the reviewer for their thoughtful and encouraging feedback. We greatly appreciate your recognition of the novelty and potential clinical relevance of our work. Your constructive suggestions have been extremely valuable in strengthening the scientific rigor and overall presentation quality of the manuscript. We have carefully addressed each of your comments in the revised version and provide detailed responses below.
- Lifestyle factors such as smoking, alcohol consumption, and physical activity, which may significantly influence FLD risk, were not considered. Please clarify this aspect in the Methods or Discussion, or explicitly acknowledge it as a limitation.
Response:
We agree with the comment. In the methods section 2.4 Cardiometabolic risk factors, we clarify that these lifestyle factors were considered for the analysis. Lines 116-118.
Lifestyle factors such as current smoking, high alcohol consumption (> 20g/d), and total physical activity (Leisure, work and exercise time), which may significantly influence FLD risk, were considered as confounding variables.
Data on physical activity and alcohol consumption were added to Table 1
- Potential selection bias related to exclusion criteria and sample selection was not discussed. Please address possible sources of selection bias and their potential impact on the findings.
Response:
Thanks a lot for this observation. We have added comments on this issue in the "Limitations" section at the end of the discussion section. Lines 417-421
Another limitation of this work is that since the sample consisted of volunteers, participants may not have represented the general population. Nevertheless, the prevalence of coronary heart disease risk factors observed is similar to that found in ENSANUT, a survey with national representation.
- Figure formatting needs improvement; for example, axis labels in the ROC curves (“1–Specificity”, “Sensitivity”) should be capitalized according to international standards.
Response:
We appreciate the reviewer’s attention to detail regarding figure formatting. As suggested, we have corrected the axis labels in all ROC curve figures to follow international conventions—specifically, “1–Specificity” and “Sensitivity” are now properly capitalized. These changes have been incorporated into the revised figures to ensure consistency and clarity.
- The ROC curves are thin, pale, and display aliasing, which compromises the clarity of key results. Please enhance figure quality by thickening lines and optimizing color schemes.
Response:
We appreciate the reviewer’s observation regarding the visual quality of the ROC curves. In response, we have regenerated all ROC figures using a high-resolution plotting software to improve their clarity and readability. Specifically, we have increased line thickness, optimized the color palette for better contrast, and eliminated aliasing artifacts. These updated figures provide a clearer visualization of the differences in predictive performance among the evaluated indices and biochemical ratios. All revised figures have been incorporated into the updated version of the manuscript.
- The best cut-off points, along with corresponding sensitivity and specificity values, are not indicated for major curves (e.g., the best model and the UA/HDL-C ratio). Including this information would strengthen the clinical relevance of the findings.
The FLI achieved the highest area under the curve (AUC = 0.755) and the highest Youden Index (0.35), indicating the best overall performance. In comparison, the UA/HDL-C ratio showed a lower AUC (0.679) and a smaller Youden Index (0.23), reflecting a more limited diagnostic utility. The optimal cut-off points were 7.39 for FLI (sensitivity = 0.80, specificity = 0.55) and 0.12 for UA/HDL-C ratio (sensitivity = 0.66, specificity = 0.57).
- The Discussion lacks a comprehensive analysis of study limitations, particularly regarding the inability of cross-sectional studies to infer causality, sample representativeness, and generalizability issues. Please add a structured limitations section.
Response:
We thank the reviewer for this important comment. We acknowledge that the current manuscript version did not include a dedicated limitations section, and we agree that a structured discussion of the study's limitations will strengthen the manuscript. Accordingly, we have added a new subsection titled "Limitations" at the end of the Discussion section to address key constraints of our study, including its cross-sectional design, issues of causality, population representativeness, and generalizability. The revised text reads as follows in lines 414-421
The study's cross-sectional nature precludes causal inferences between the UA/HDL-C ratio and the development of fatty liver disease. Longitudinal studies are needed to confirm whether alterations in this ratio precede hepatic steatosis or result from it. Another limitation of this work is that since the sample consisted of volunteers, participants may not have represented the general population. Nevertheless, the prevalence of coronary heart disease risk factors observed is similar to that found in the ENSANUT, a survey with national representation [64].
- The manuscript does not sufficiently compare the UA/HDL-C ratio with traditional composite indices (e.g., FLI, LAP) in terms of predictive ability and practical application. A more detailed discussion highlighting their complementary roles would be beneficial.
Response:
We thank the reviewer for this insightful comment. We agree that a more detailed comparative discussion of the UA/HDL-C ratio versus traditional composite indices such as FLI and LAP will provide additional clarity regarding their predictive strengths and practical applications. In response, we have expanded the Discussion section to better emphasize the complementary roles of the UA/HDL-C ratio and established indices in the context of FLD risk assessment. The revised text is now included in the Discussion as follows in lines 314-332:
While our results confirm that the UA/HDL-C ratio has a lower area under the curve (AUC) compared to traditional composite indices such as the Fatty Liver Index (FLI), NAFLD score, and Hepatic Steatosis Index (HSI), it is important to consider the clinical context and utility of each approach. Composite indices integrate multiple anthropometric and biochemical parameters, enhancing diagnostic performance but requiring broader data availability and more complex calculations. In contrast, the UA/HDL-C ratio is derived from only two routinely measured laboratory parameters, making it highly accessible, cost-effective, and easy to implement in clinical settings, particularly in resource-limited environments.
Moreover, our findings suggest that while composite indices may be more robust in overall predictive capacity, the UA/HDL-C ratio provides complementary information that reflects specific metabolic disturbances—namely, hyperuricemia and hypoalphalipoproteinemia—both of which are independently linked to insulin resistance and hepatic steatosis. The ratio may thus serve as a rapid, first-line screening tool, especially in primary care or epidemiological studies where full biochemical and anthropometric profiles are unavailable. Furthermore, sex-stratified analyses revealed that the UA/HDL-C ratio maintains strong performance in men, highlighting its potential as a sex-sensitive marker. Together, these findings support the combined use of both approaches: composite indices for comprehensive evaluation, and the UA/HDL-C ratio for initial screening or follow-up monitoring.
In lines 345-356
Interestingly, the ALT/AST ratio, traditionally used as a liver injury marker, performed comparably but not superior to the UA/HDL-C ratio in either sex. This could reflect the fact that ALT and AST elevations typically indicate more advanced hepatic damage, whereas the UA/HDL-C ratio may capture earlier metabolic alterations linked to steatosis onset. LDL-C and LDL-C/HDL-C ratios, despite their widespread use in cardiovascular risk prediction, showed weak to moderate discriminative power for FLD, reinforcing the notion that hepatic steatosis is more tightly linked to triglyceride-rich lipoprotein metabolism and insulin resistance than to isolated elevations in LDL-C.
Collectively, these results emphasize the potential utility of the UA/HDL-C ratio as a clinically accessible marker that integrates both inflammatory and lipid-related pathways relevant to FLD, with sex-specific implications. Its performance, particularly in men, suggests it may be a valuable addition to routine metabolic screening in high-risk populations.
Reviewer 3 Report
Comments and Suggestions for Authors
This study provides valuable evidence that UA/HDL-C is a feasible FLD biomarker, particularly in resource-limited settings. With revisions to address methodological nuances and contextualize sex/ethnic variability, it could influence clinical practice. I think there are only a few minor problems that the author can fix and publish. Including:
- Define abbreviations at first use (e.g., L/SAR in Methods).
- Fix minor errors (e.g., "Remains asymptomatic" → "It remains asymptomatic").
- Improve resolution; ensure font consistency in ROC curves.
Author Response
Dear Reviewer 3
Thank you for taking the time to review and comment on our manuscript,
Is UA/HDL-C a Reliable Surrogate Marker for Fatty Liver? A Comparative Evaluation with Metabolic Scores in a Mexican Population. (Diagnostics 3633690).
We found the advice constructive and incorporated many suggestions into our reviewed version.
We have responded to each comment individually and would like to draw your attention to the changes in the manuscript, highlighted in red.
Thank you again for your thoughtful comments.
Sincerely,
Dr. Guillermo C. Cardoso-Saldaña.
Reviewer 3
- Define abbreviations at first use (e.g., L/SAR in Methods).
Response:
Thanks for this suggestion. We highlight the diagnostic method used to define fatty liver disease to clarify this point.
Non-alcoholic fatty liver disease was diagnosed using the Liver/Spleen Attenuation Ratio (L/SAR), with a ratio of <1.0 indicating the presence of FLD, as described by Longo R. et al. [34]. Line 112
- Fix minor errors (e.g., "Remains asymptomatic" → "It remains asymptomatic").
Response:
Thank you so much for this observation. We fixed this error as you suggested.
Line 18
- Improve resolution; ensure font consistency in ROC curves.
Response:
We appreciate the reviewer’s observation regarding the visual quality of the ROC curves. In response, we have regenerated all ROC figures using a high-resolution plotting software to improve their clarity and readability. Specifically, we have increased line thickness, optimized the color palette for better contrast, and eliminated aliasing artifacts. These updated figures provide a clearer visualization of the differences in predictive performance among the evaluated indices and biochemical ratios. All revised figures have been incorporated into the updated version of the manuscript.
Round 2
Reviewer 2 Report
Comments and Suggestions for Authors
1. Issues with Language Clarity and Redundancy
Several parts of the manuscript contain long sentences and repetitive expressions, which reduce readability and weaken the logical flow. For example, the advantages of the UA/HDL-C ratio such as its simplicity, cost-effectiveness, and accessibility are emphasized repeatedly using similar wording.
Original examples:
“The UA/HDL-C ratio is derived from only two routinely measured laboratory parameters, making it highly accessible, cost-effective, and easy to implement in clinical settings, particularly in resource-limited environments.”
“The UA/HDL-C ratio provides a simplified and accessible alternative for rapid screening.”
Such expressions appear repeatedly throughout the manuscript and should be condensed into a single, clear statement to avoid redundancy.
There are also minor errors in spelling and terminology. For example:
Original error:
“to evaluate and compare the predictived capacity of various indices for FLD”
Correction:
“predictive capacity”
Suggestion: A comprehensive language edit is recommended to remove redundancy, correct minor errors, and enhance the overall clarity and scientific tone of the manuscript.
2. Mechanistic Discussion is Rich but Overloaded
The mechanistic section linking the UA/HDL-C ratio to FLD pathogenesis is detailed but suffers from excessive information stacking and lack of structural clarity. Multiple signaling pathways and molecular events are listed in succession (e.g., JNK/NF-κB, INSR, SREBP, KEGG), making it difficult for the reader to follow the main narrative.
Original example:
“Uric acid, a byproduct of purine metabolism, actively contributes to FLD development by inducing oxidative stress, promoting mitochondrial dysfunction, activating the JNK1-NF-κB pathway, and driving this repeated inflammation-related signal, which in turn stimulates hepatic de novo lipogenesis by upregulating transcription factors such as SREBP-1c and ChREBP.”
Suggestion:
Consider presenting these mechanistic pathways using a figure or schematic diagram, and structuring the text so that each paragraph focuses on one major pathway or biological process. This will improve clarity and help readers understand the relevance of each mechanism to the central hypothesis.
3. Overstated Conclusion and Lack of Highlighted Contributions
The conclusion currently includes statements that may be too strong given the observational nature of the study. For example, using phrases like “is a useful marker” or “is a relevant metabolic indicator” implies a level of certainty that may not be appropriate for a cross-sectional analysis. It is better to use more cautious and scientifically accurate wording.
Suggestion:
Replace assertive phrases with more cautious alternatives, such as:
“Our findings suggest that the UA/HDL-C ratio may serve as a useful metabolic indicator for FLD…”
Author Response
Response to Reviewer Round 2
- Issues with Language Clarity and Redundancy
Several parts of the manuscript contain long sentences and repetitive expressions, which reduce readability and weaken the logical flow. For example, the advantages of the AU/HDL-C ratio such as its simplicity, cost-effectiveness, and accessibility are emphasized repeatedly using similar wording.
The original sentence:
“The UA/HDL-C ratio is derived from only two routinely measured laboratory parameters, making it highly accessible, cost-effective, and easy to implement in clinical settings, particularly in resource-limited environments.”
Response:
To avoid this issue, the long sentences and repetitive expressions were deleted
as the Reviewer suggested.
“In contrast, the UA/HDL-C ratio provides a simplified and accessible alternative for rapid screening.” Pag. 10 lines 307-310
“The ratio may thus serve as a rapid, first-line screening tool, especially in primary care or epidemiological studies where complete biochemical and anthropometric profiles are unavailable.” Wass eliminated. Pag. 10 line 315
There are also minor errors in spelling and terminology.
Response:
We review all the manuscript for spelling errors and terminology.
- Mechanistic Discussion is Rich but Overloaded
The mechanistic section linking the UA/HDL-C ratio to FLD pathogenesis is detailed but suffers from excessive information stacking and lack of structural clarity. Multiple signaling pathways and molecular events are listed in succession (e.g., JNK/NF-κB, INSR, SREBP, KEGG), making it difficult for the reader to follow the main narrative.
Original example:
“Uric acid, a byproduct of purine metabolism, actively contributes to FLD development by inducing oxidative stress, promoting mitochondrial dysfunction, activating the JNK1-NF-κB pathway, and driving this repeated inflammation-related signal, which in turn stimulates hepatic de novo lipogenesis by upregulating transcription factors such as SREBP-1c and ChREBP.”
Suggestion:
Consider presenting these mechanistic pathways using a figure or schematic diagram, and structuring the text so that each paragraph focuses on one major pathway or biological process. This will improve clarity and help readers understand the relevance of each mechanism to the central hypothesis.
Response:
We thank the reviewer for this valuable comment. In response, we have carefully revised the mechanistic section to improve its structure, clarity, and readability. Specifically, we reorganized the content to first present the clinical correlation findings between the UA/HDL-C ratio and relevant metabolic markers (e.g., glucose, ALT, HOMA-IR, triglycerides, and waist circumference), followed by a distinct and focused description of the in silico protein–protein interaction analysis.
Additionally, as suggested, we added a schematic figure that summarizes the key molecular mechanisms linking hyperuricemia and hypoalphalipoproteinemia to FLD progression. This figure is provided as Supplementary Data 2; however, we have also embedded the figure within the revised discussion section of the main text in case the reviewer or editors consider it more appropriate to include it directly in the manuscript body rather than as supplementary material.
We believe these changes greatly enhance the clarity and accessibility of the mechanistic discussion, aligning it more effectively with the central hypothesis of the study. We sincerely appreciate the reviewer’s suggestion, which significantly improved this section.
- Overstated Conclusion and Lack of Highlighted Contributions
The conclusion currently includes statements that may be too strong given the observational nature of the study. For example, using phrases like “is a useful marker” or “is a relevant metabolic indicator” implies a level of certainty that may not be appropriate for a cross-sectional analysis. It is better to use more cautious and scientifically accurate wording.
Suggestion:
Replace assertive phrases with more cautious alternatives, such as:
“Our findings suggest that the UA/HDL-C ratio may serve as a useful metabolic indicator for FLD…”
Response:
In conclusions, the strong statements such as Relevant, useful, strong, and it may serve were changed, as reviewer sugested. Pag. 13, line 417 -421
Our findings suggest that the UA/HDL-C ratio may be a metabolic indicator for FLD, associated with biochemical disturbances in hepatic lipid accumulation. While its predictive capacity is lower than traditional indices, its association with metabolic dysfunction suggests it could be a helpful marker. However, further studies are warranted to explore its potential utility in risk stratification and early detection of FLD.
Round 3
Reviewer 2 Report
Comments and Suggestions for Authors
This manuscript presents a novel and well-designed study with rigorous data analysis and convincing results. The overall structure is clear, and the language is appropriate. I believe the manuscript is suitable for publication.